Effects of sex and site on amino acid metabolism enzyme gene expression and activity in rat white adipose tissue

Arriarán Sofía 1
Agnelli Silvia 1
Remesar Xavier 1 2 3
Fernández-López José Antonio 1 2 3
Alemany Marià 1 2 3 malemany@ub.edu
1 Department of Nutrition & Food Science, University of Barcelona, Faculty of Biology , Barcelona , Spain
2 Institute of Biomedicine, University of Barcelona , Barcelona , Spain
3 CIBER OBN , Barcelona , Spain
Barnett Matthew
Electronic publication date: 2015 Nov 10
Publication date: 2015
Volume: 3
Electronic Location ID: e1399
Received 2015 Jul 28; Accepted 2015 Oct 21
Copyright: © 2015 Arriarán et al.
Copyright year: 2015
Copyright holder: Arriarán et al.
License: This is an open access article distributed under the terms of the Creative Commons Attribution License, which permits unrestricted use, distribution, reproduction and adaptation in any medium and for any purpose provided that it is properly attributed. For attribution, the original author(s), title, publication source (PeerJ) and either DOI or URL of the article must be cited.
License URL: https://creativecommons.org/licenses/by/4.0/

Keywords: Adipose tissue, Urea cycle, Ammonium, Citrulline, Urea, Adipose organ, Lipogenesis

Funding: Plan Nacional de Ciencia y Tecnología de los Alimentos AGL-2011-23635 Plan Nacional de Investigación en Biomedicina SAF2012-34895 CIBER-OBN Research Web This work was funded by the Plan Nacional de Ciencia y Tecnología de los Alimentos (AGL-2011-23635) of the Government of Spain and Plan Nacional de Investigación en Biomedicina (SAF2012-34895) of the Government of Spain. CIBER-OBN Research Web, Barcelona, Spain. Silvia Agnelli was the recipient of a Leonardo da Vinci fellowship. Sofía Arriarán had a predoctoral fellowship of the Catalan Government. The funders had no role in study design, data collection and analysis, decision to publish, or preparation of the manuscript.

==============================
Background and Objectives. White adipose tissue (WAT) shows marked sex- and diet-dependent differences. However, our metabolic knowledge of WAT, especially on amino acid metabolism, is considerably limited. In the present study, we compared the influence of sex on the amino acid metabolism profile of the four main WAT sites, focused on the paths related to ammonium handling and the urea cycle, as a way to estimate the extent of WAT implication on body amino-nitrogen metabolism.

Experimental Design. Adult female and male rats were maintained, undisturbed, under standard conditions for one month. After killing them under isoflurane anesthesia. WAT sites were dissected and weighed. Subcutaneous, perigonadal, retroperitoneal and mesenteric WAT were analyzed for amino acid metabolism gene expression and enzyme activities.

Results. There was a considerable stability of the urea cycle activities and expressions, irrespective of sex, and with only limited influence of site. Urea cycle was more resilient to change than other site-specialized metabolic pathways. The control of WAT urea cycle was probably related to the provision of arginine/citrulline, as deduced from the enzyme activity profiles. These data support a generalized role of WAT in overall amino-N handling. In contrast, sex markedly affected WAT ammonium-centered amino acid metabolism in a site-related way, with relatively higher emphasis in males’ subcutaneous WAT.

Conclusions. We found that WAT has an active amino acid metabolism. Its gene expressions were lower than those of glucose-lipid interactions, but the differences were quantitatively less important than usually reported. The effects of sex on urea cycle enzymes expression and activity were limited, in contrast with the wider variations observed in other metabolic pathways. The results agree with a centralized control of urea cycle operation affecting the adipose organ as a whole.

Introduction

The influence of sex on adipose tissue distribution and function, and its implication in metabolic syndrome has been known for a long time (Mayes & Watson, 2004). The protective effects of estrogen on adipose tissue activity (D’Eon et al., 2005), and limitation of its hypertrophic growth (Kumar et al., 2012) and inflammation (Stubbins et al., 2012) are key sex-related factors, which contribute to limit the disorders elicited by metabolic syndrome (Antonio et al., 2015). The distribution of fat in gynoid and android shapes of adult humans is a consequence of the close interrelationship of adipose tissue with androgens and estrogens (Kotani et al., 1994), modulated by their different response to glucocorticoids in the aftermath of a tissue defensive response against excess nutrient loads (Alemany, 2012a).

There are clear differences between females and males in site distribution and metabolic responses (Porter et al., 2004; Demerath et al., 2007), but most studies on adipose tissue are limited to a single site or isolated cells, and are usually focused on the responses to inflammation (Revelo et al., 2014).

The extensive metabolic capability of white adipose tissue (WAT) show a remarkable uniformity in metabolic function and overall regulation (Carmean, Cohen & Brady, 2014; Romero et al., 2014). As a consequence, the assumed main function of WAT (i.e., triacylglycerol storage as energy reserve) (Galic, Oakhill & Steinberg, 2010; Romacho et al., 2014) is been reconsidered because of the multiple functions of this unique disperse organ (Eringa, Bakker & Van Hinsbergh, 2012; Ferrante, 2013; Giordano et al., 2014). However, the level of knowledge of WAT metabolism, other than the control of lipid synthesis and storage, remains remarkably insufficient, constituting a handicap for interpretation of its physiological role (Jensen, 2007).

WAT contains a complete urea cycle, as shown in the present study, which is, probably implicated in the extra-splanchnic production of citrulline, a critical factor for muscle function (Ventura et al., 2013) and inter-organ 2-amino-N transport and utilization. However, WAT is also a massive producer of 3C fragments, such as lactate (Arriarán et al., 2015), but including alanine (Snell & Duff, 1977). WAT is also a net exporter of glutamine (Kowalski & Watford, 1994), and can use branched-chain amino acids for energy and lipogenesis (Herman et al., 2010). The large combined organ size, variety of known amino acid metabolic pathways and diverse physiological functions, hint at WAT as a potentially important site for peripheral amino acid metabolism. The information available is scant, we found only a couple of earlier studies (López-Soriano & Alemany, 1986; Kowalski, Wu & Watford, 1997); this is a serious limitation for a full understanding of whether amino acids should be also included in the well-established role of WAT in the management of energy, from glucose and lipids.

The little we know of WAT role in amino acid metabolism is further limited by our almost nil understanding of the role sex plays on WAT metabolism. In general terms, androgens favor protein deposition (Griggs et al., 1989), and males tend to consume spontaneously more protein than females (Radcliffe & Webster, 1978); on the other hand, estrogens lower body weight (Bryzgalova et al., 2008), in spite of females (women) having—normally—a higher body fat percentage than males (men). Young women are more resistant to obesity than men (Meyer et al., 2011); however, after menopause, this estrogenic protection wanes (Cagnacci et al., 2007).

In this study, we intended to determine whether the gross differences in WAT distribution and its resilience to change had a robust biochemical basis. Thus, we analyzed whether the WAT urea cycle and related amino acid catabolic processes of rats showed sex-modulated differences. To obtain a wider picture we studied the four main (largest) WAT sites in parallel, and we included in the analysis (for comparison) a number of gene expressions involved in the control of WAT lipogenesis from glucose and lipolysis.

Materials and Methods

Experimental design and animal handling

All animal handling procedures and the experimental setup were in accordance with the animal handling guidelines of the corresponding European and Catalan Authorities. The Committee on Animal Experimentation of the University of Barcelona specifically authorized the procedures used in the present study (DMAH-5483).

The experimental setup consisted on keeping two groups of undisturbed rats (female and male) under standard conditions for four weeks, in order to limit the influence of factors other than sex on the parameters analyzed.

Nine week old female and male Wistar rats (Harlan Laboratory Models, Sant Feliu de Codines, Spain) were used. The rats (N = 6 per group) were housed in pairs (same sex) in solid-bottom cages with wood shards for bedding. They had free access to water and ate normal rat chow (type 2014, Harlan). The rats were kept in a controlled environment (lights on from 08:00 to 20:00; 21.5–22.5 °C; 50–60% humidity) for one month.

The rats, without dietary manipulation, were killed, under isoflurane anesthesia, at the beginning of a light cycle (08:30–10:00), by aortic exsanguination, using dry-heparinized syringes; then, they were rapidly dissected, taking samples of WAT sites: mesenteric (ME), perigonadal (epididymal in males, periovaric in females, PG), retroperitoneal (RP) and subcutaneous (inguinal fat pads, SC). The samples were blotted and frozen with liquid nitrogen; after weighing, they were ground under liquid nitrogen and stored at −80 °C until processed. Later, the dissection of the rats continued, extracting the remaining WAT in ME, EP and RP sites; the rats were skinned, and the whole subcutaneous WAT was dissected. The weights of the recovered WAT were computed only to establish the total mass of each WAT site.

Blood plasma parameters

The blood obtained from the aorta was centrifuged to obtain plasma, which was frozen and kept at −80 °C until processed. Plasma samples were used to measure glucose (kit #11504), triacylglycerols (kit #11828), total cholesterol (kit #11505) and urea (kit # 11537), all from Biosystems, Barcelona Spain. Lactate was measured with another kit (ref. #1001330; Spinreac, Sant Esteve de Bas, Spain). Amino acids were analysed individually using an amino acid analyser (Pharmacia-LKB-Alpha-plus, Uppsala, Sweden) from plasma samples deproteinized with acetone (Arola, Herrera & Alemany, 1977). Since the method used did not provide fair analyses for glutamine (Gowda, Gowda & Raftery, 2015) and other amino acids (Trp, Cys, Asn), we decided to present only the partial sum of the other amino acids as a single indicative value.

Preparation of tissue homogenates

Frozen tissue samples were homogenized, using a tissue disruptor (Ultraturrax IKA-T10, Ika Werke, Staufen, Germany), in 5 volumes of chilled 70 mM hepes buffer pH 7.4 containing 1 mM dithiothreitol (Sigma, St Louis MO USA), 50 mM KCl, 1 g/L Triton X-100 (Sigma) and 1 g/L lipid-free bovine serum albumin (Sigma). In homogenates to be used for carbamoyl-P synthase 2 estimation, the concentration of Triton X-100 was halved to decrease foaming. The homogenates were centrifuged for 10 min at 5,000 × g; the floating fat layers and gross debris precipitates were discarded. The clean homogenates were kept on ice, and used for enzymatic analyses within 2 h of their preparation.

Tissue protein content was estimated with the Lowry method (Lowry et al., 1951). After development of color, fat droplet suspension-generated turbidity was eliminated with the addition of small amounts of finely powdered solid MgO before reading the absorbance. In the measurements of homogenate protein content, homogenization buffer (which contained albumin) was used as blank. Enzyme activities were expressed in nkat/g protein.

Enzyme activity analyses

Carbamoyl-P synthase was estimated from the incorporation of 14C-bicarbonate (Perkin Elmer, Bad Neuheim, Germany) into carbamoyl-P using a method previously described by us (Arriarán et al., 2012). No significant carbamoyl-P synthase 1 activity was detected (and its gene was not expressed, either, in WAT). Thus, only carbamoyl-P synthase 2 was measured.

All other enzyme activities (ornithine carbamoyl-transferase, arginino-succinate synthase, arginino-succinate lyase and arginases 1 and 2) were estimated following recently developed methods, which are presented in detail in Supplemental Information 1 both to justify their adequacy and to allow others to employ a methodology developed for adipose tissue.

Gene expression analysis

Total tissue RNA was extracted from frozen tissue samples using the Tripure reagent (Roche Applied Science, Indianapolis IN USA), and was quantified in a ND-100 spectrophotometer (Nanodrop Technologies, Wilmington DE USA). These data were also used to determine the total RNA content of the tissue (per g of tissue weight or g of protein) in order to establish comparisons between the quantitative importance of gene expressions. RNA samples were reverse transcribed using the MMLV reverse transcriptase (Promega, Madison, WI USA) system and oligo-dT primers.

Real-time PCR (RT-PCR) amplification was carried out using 10 µL amplification mixtures containing Power SYBR Green PCR Master Mix (Applied Biosystems, Foster City, CA USA), 4 ng of reverse-transcribed RNA and 150 nmol of primers. Reactions were run on an ABI PRISM 7900 HT detection system (Applied Biosystems) using a fluorescent threshold manually set to 0.15 for all runs.

A semi-quantitative approach for the estimation of the concentration of specific gene mRNAs per unit of tissue/RNA or protein weight was used (Romero et al., 2007). Rplp0 was the charge control gene (Eagni et al., 2013). We expressed the data primarily as the number of transcript copies per gram of protein in order to obtain comparable data between the groups. The genes analyzed and a list of primers used is presented in Table 1.

Table 1 Primer sequences used in the analysis of WAT gene expressions.

	Protein	Gene	EC	Primer sequence 5′ > 3′	Primer sequence 3′ > 5′	bp	
CPS2	Glutamine-dependent carbamoyl-phosphate synthase	Cad	6.3.5.5	AGTTGGAGGAGGAGGCTGAG	ATTGATGGACAGGTGCTGGT	90	
OTC	Ornithine carbamoyl transferase	Otc	2.1.3.3	CTTGGGCGTGAATGAAAGTC	ATTGGGATGGTTGCTTCCT	126	
ASS	Arginino-succinate synthase	Ass1	6.3.4.5	CAAAGATGGCACTACCCACA	GTTCTCCACGATGTCAATGC	100	
ASL	Arginino-succinate lyase	Asl	4.3.2.1	CCGACCTTGCCTACTACCTG	GAGAGCCACCCCTTTCATCT	104	
ARG1	Arginase-1	Arg1	3.5.3.1	GCAGAGACCCAGAAGAATGG	GTGAGCATCCACCCAAATG	126	
ARG2	Arginase-2	Arg2	3.5.3.1	GCAGCCTCTTTCCTTTCTCA	CCACATCTCGTAAGCCAATG	122	
NAGS	N-acetyl-glutamate synthase	Nags	2.3.1.1	GCAGCCCACCAAAATCAT	CAGGTTCACATTGCTCAGGA	82	
eNOS	Nitric oxide synthase, endothelial	Nos3	1.14.13.39	CAAGTCCTCACCGCCTTTT	GACATCACCGCAGACAAACA	138	
GS	Glutamine synthetase	Glul	6.3.1.2	AACCCTCACGCCAGCATA	CTGCGATGTTTTCCTCTCG	148	
Gase	Glutaminase kidney isoform, mitochondrial	Gls	3.5.1.2	CCGAAGGTTTGCTCTGTCA	AGGGCTGTTCTGGAGTCGTA	63	
GDH1	Glutamate dehydrogenase 1, mitochondrial	Glud1	1.4.1.3	GGACAGAATATCGGGTGCAT	TCAGGTCCAATCCCAGGTTA	122	
GCS	Glycine cleavage system H protein, mitochondrial	Gcsh	–	AAGCACGAATGGGTAACAGC	TCCAAAGCACCAAACTCCTC	146	
AMPD	AMP deaminase 2	Ampd2	3.5.4.6	CGGCTTCTCTCACAAGGTG	CGGATGTCGTTACCCTCAG	78	
AlaT1	Alanine aminotransferase 1	Gpt	2.6.1.2	GTATTCCACGCAGCAGGAG	CACATAGCCACCACGAAACC	85	
AlaT2	Alanine aminotransferase 2	Gpt2	2.6.1.2	CATTCCCTCGGATTCTCATC	GCCTTCTCGCTGTCCAAA	146	
BCT1	Branched-chain-amino-acid aminotransferase, cytosolic	Bcat1	2.6.1.42	TGCCCAGTTGCCAGTATTC	CAGTGTCCATTCGCTCTTGA	138	
BCT2	Branched-chain-amino-acid aminotransferase, mitochondrial	Bcat2	2.6.1.42	AGTCTTCGGCTCAGGCACT	ATGGTAGGAATGTGGAGTTGCT	84	
GLUT4	Solute carrier family 2 (facilitated glucose transporter), member 4	Glut4	–	CACAATGAACCAGGGGATGG	CTTGATGACGGTGGCTCTGC	127	
HK	Hexokinase-2	Hk2	2.7.1.1	ATTCACCACGGCAACCACAT	GGACAAAGGGATTCAAGGCATC	113	
G6PDH	Glucose-6-phosphate 1-dehydrogenase	G6pdx	1.1.1.49	GACTGTGGGCAAGCTCCTCAA	GCTAGTGTGGCTATGGGCAGGT	77	
ME	NADP-dependent malic enzyme	Me1	1.1.1.40	TTCCTACGTGTTCCCTGGAG	GGCCTTCTTGCAGGTGTTTA	131	
PDHK2	Pyruvate dehydrogenase kinase 2, mitochondrial	Pdk2	2.7.11.2	TCACTCTCCCTCCCATCAA	CGCCTCGGTCACTCATTT	75	
PDHK4	Pyruvate dehydrogenase [acetyl transferring] kinase 4, mitochondrial	Pdk4	2.7.11.2	GTCAGGCTATGGGACAGATGC	TTGGGATACACCAGTCATCAGC	137	
CATPL	ATP citrate lyase	Acly	2.3.3.8	GACCAGAAGGGCGTGACCAT	GTTGTCCAGCATCCCACCAGT	96	
ACoAC	Acetyl-CoA carboxylase 1	Acaca	6.4.1.2	AGGAAGATGGTGTCCGCTCTG	GGGGAGATGTGCTGGGTCAT	145	
FAS	Fatty acid synthase	Fasn	2.3.1.85	CTTGGGTGCCGATTACAACC	GCCCTCCCGTACACTCACTC	163	
PCATl	Carnitine palmitoyltransferase 1, liver isoform	Cpt1a	2.3.1.21	CCGCTCATGGTCAACAGCA	CAGCAGTATGGCGTGGATGG	105	
PCATm	Carnitine palmitoyltransferase 2, mitochondrial	Cpt2	2.3.1.21	TGCTTGACGGATGTGGTTCC	GTGCTGGAGGTGGCTTTGGT	152	
ACADH	Long-chain acyl-CoA dehydrogenase, mitochondrial	Acadl	1.3.8.8	ATGCCAAAAGGTCTGGGAGT	TCGACCAAAAAGAGGCTAATG	148	
ATL	Adipose triacylglycerol lipase	Atgl	3.1.1.3	CGGTGGATGAAGGAGCAGACA	TGGCACAGACGGCAGAGACT	138	
HSL	Hormone-sensitive lipase	Lipe	3.1.1.79	CCCATAAGACCCCATTGCCTG	CTGCCTCAGACACACTCCTG	94	
LPL	Lipoprotein lipase	Lpl	3.1.1.34	GAAGGGGCTTGGAGATGTGG	TGCCTTGCTGGGGTTTTCTT	103	
	60S acidic ribosomal protein 0 (housekeeping gene)	Rplp0	–	GAGCCAGCGAAGCCACACT	GATCAGCCCGAAGGAGAAGG	62	

Figure 1 depicts a scheme of the relationships between the amino acid metabolism-related enzymes which gene expressions have been analyzed in this study. This Figure also shows acronyms or abbreviations of the names of the enzyme-genes used in Figs. 2 and 3.

Figure 1 Scheme of the core of amino acid metabolism in WAT: urea cycle and ammonium handling.

The abbreviations (marked in red) of the enzymes involved in the pathways depicted are the same described in Table 1 and throughout this study.

Sex differences in gene expression

The ample variability of cell volume, blood flow, innervation, size of fat deposits etc. of WAT poses additional problems for comparison between different anatomical (i.e., site), physiological (i.e., sex, diet) and pathological (i.e., obesity) situations (Caspar-Bauguil et al., 2005; Prunet-Marcassus et al., 2006; Gil et al., 2011). The variability of lipid reserves may convert in irrelevant comparisons based on weight; the use of DNA or cell number is a better approach, but the multiple types of cells coexisting in WAT (and their widely variable numbers) may also alter direct comparisons. We used as basic comparison the protein content, largely because it has been the choice reference for enzyme activity and, by extension to gene expressions and their possible interactions. However, probably a better way to measure changes in functional activity may be the analysis of mRNA production. These changes do not parallel those of weight, protein content or cell numbers, but are a fair index of the relative importance of translation of the genes involved with respect to total protein synthesis. In fact, their use is complementary of the analysis of enzyme activity-gene expression referred to protein weight, since it includes a new variable: metabolic transcendence for the cell of the synthesis of the corresponding mRNAs. We calculated the relationships of gene expressions to total tissue RNA (a crude approximation to mRNA) only to compare the specific effect of sex on a gene expression in a given site. Any significant deviation on the proportion of a gene expression with respect to the whole RNA mass may imply a differential modulation of this expression. We included these additional data to provide further insight into the ways and means of manifestation of sex-related differences.

Statistics

Student’s t test (unpaired) and two-way ANOVA comparisons between groups (using the post-hoc Tuckey test), correlations and curve fitting (including Vi estimations) were analyzed with the Prism 5 program (GraphPad Software, San Diego CA USA). Data were presented as mean ± sem, and a limit of significance of P < 0.05 was used throughout.

Results

Basic parameters

Table 2 shows the body and main adipose tissue sites weights of undisturbed female and male animals. When aged 13 weeks, female rats weighed about 62% of their male counterparts. The males accumulated more fat than females, both in individual sites and as a whole. However, the sum of the four sites analyzed showed almost identical proportions vs. body weight, c. 8%. However, there were sex-related individual site differences in relative size expressed as percentage of body weight. There were also differences in total protein and RNA proportions (per g of fresh tissue) between the different sites, but there were no global effects attributable to the variable “sex”.

Table 2 Body and WAT site weight and composition of adult male and female Wistar rats.

The data correspond to the mean ± sem of 6 different animals. Statistical significance of the differences between groups was established with a 2-way anova; post-hoc Tuckey test: an asterisk * represents P < 0.05 differences between sex groups. Comparison of differences between the sums of sites was done using the Student’s t test.

Parameter	Unit	Site	Male	Female	p site	p sex	
Body weight	g	–	373 ± 6.1	232 ± 8.2	–	<0.0001	
WAT weight	g	SC	12.2 ± 0.20*	7.02 ± 0.25	<0.0001	<0.0001	
ME	4.94 ± 0.49*	3.92 ± 0.33	
PE	7.34 ± 0.64*	4.83 ± 0.39	
RE	6.29 ± 0.79*	2.79 ± 0.35	
Σ WAT	30.8 ± 1.7	18.6 ± 0.93	–	<0.0001	
% BW	SC	3.28 ± 0.05	3.04 ± 0.11	<0.0001	NS	
ME	1.33 ± 0.18	1.69 ± 0.13	
PE	1.97 ± 0.13	2.10 ± 0.21	
RE	1.69 ± 0.22	1.22 ± 0.17	
Σ WAT	8.26 ± 0.47	8.05 ± 0.52	–	NS	
Protein	mg/g	SC	63.1 ± 11.6	51.8 ± 3.3	<0.0001	NS	
ME	74.2 ± 7.4	84.2 ± 2.6	
PE	44.3 ± 1.6	54.4 ± 2.4	
RE	65.1 ± 6.3	62.9 ± 4.7	
RNA	μg/g	SC	248 ± 51	219 ± 19	<0.0001	NS	
ME	880 ± 84	793 ± 88	
PE	94.3 ± 6.0	119 ± 10	
RE	48.8 ± 4.1	78.4 ± 4.1	
Notes.

% BW Percentage of body weight

The main plasma parameters studied are presented in Table 3. Plasma glucose levels were higher in males than in females. However, these data were influenced by isoflurane anesthesia (Zardooz et al., 2010), and are presented only as a general indication of normalcy. No differences were observed for lactate and total cholesterol. Triacylglycerol levels were significantly higher in females (albeit in the limit of statistical significance). Both plasma urea and the partial sum of amino acids were also higher in female than in male rats.

Table 3 Main energy plasma parameters of adult female and male Wistar rats.

The data correspond to the mean + sem of 6 different animals. Statistical significance of the differences between groups was established with the unpaired Student’s t test.

Parameter	Units	Male	Female	P sex	
Glucose	mM	10.20 ± 0.42	8.64 ± 0.34	0.0169	
Lactate	mM	3.10 ± 0.29	3.78 ± 0.24	NS	
Total cholesterol	mM	1.97 ± 0.07	1.98 ± 0.16	NS	
Triacylglycerols	mM	1.50 ± 0.06	1.69 ± 0.06	0.0491	
Urea	mM	3.90 ± 0.17	5.13 ± 0.25	0.0029	
Amino acidsa	mM	3.34 ± 0.08	3.96 ± 0.18	0.0104	
Notes.

a This value does not include Gln, Asn, Trp and Cys.

Urea cycle enzymes

Figure 2 depicts urea-cycle enzyme activities and the expression of their corresponding genes in four main WAT sites of male and female rats. In both cases, activity and gene expression, the data were presented per g of tissue protein. The data are displayed on a log scale to allow a visual comparison of the site patterns of enzyme activities and expressions. The data used in this representation are also tabulated in numeric form in Tables S1 and S2. There was a considerable coincidence in the patterns of enzyme activity distribution (and male–female similarities) in enzyme activities for all four sites. This pattern was not paralleled by that of the corresponding gene expression data, which also showed considerable uniformity in their patterns across the WAT sites. The statistical analysis of the data in Fig. 2 showed significant differences for “site” for all enzyme expressions except for arginino-succinate synthase. The site-related differences in enzyme activities, however, were limited to arginino-succinate lyase and carbamoyl-P synthase.

Figure 2 Urea cycle enzyme activities and expressions of their coding genes in four WAT sites of female and male rats.

All data are the mean ± sem of 6 animals, and are presented in a log scale. The numerical data are shown in Tables S1 and S2. Panels in column (A): enzyme activities, red (intense colour) columns correspond to males and orange (light colour) corresponds to female rats. (B): gene expressions, blue (intense colour) columns represent the males, and green (light colour) represents the females. CPS2, carbamoyl-P synthase 2; OTC, ornithine carbamoyl-transferase; ASS, arginino-succinate synthase; ASL, arginino-succinate lyase; ARG, arginase. Statistical analysis (2-way anova) of the differences between groups. Activity: there were no significant differences for “sex”; CPS2 and ASL showed P < 0.0001 for “site”. Expression: only CPS2 showed a significant (P = 0.0002) for “sex”: there were significant differences for “site” in CPS2 and ASL (P < 0.0001), OTC (P = 0.0081), ARG1 + 2 (P < 0.0001); ASS showed no significant differences. The application of post-hoc Tuckey test between male/female pairs are shown in the Figure as red stars.

Subcutaneous WAT showed more differences between sexes than other locations, affecting carbamoyl-P synthase 2 (both activity and gene expression) and arginino-succinate lyase (only activity). Arginino-succinate synthase activity showed differences between females and males in mesenteric WAT, and its higher expression was observed in periovaric WAT.

Other amino acid metabolism-related gene expressions

Figure 3 shows the gene expressions of the non-urea cycle enzymes presented in Fig. 1, as well as differentiated arginases 1 and 2, which were combined in Fig. 2. The data are depicted also on a log scale to facilitate pattern comparison; the corresponding numerical results are shown in Table S3.

Figure 3 Expression of genes coding for enzymes of amino acid metabolism in WAT sites of male and female rats.

All data are the mean ± sem of 6 animals, and are presented in a log scale. The numerical data are shown in Table S3. Blue (dark colour) columns represent the males, and green (light colour) represents the females. NAGS, N-acetyl glutamate synthase; Ase1, arginase 1; Ase 2, Arginase 2; eNOS, endothelial nitric oxide synthase; GS, glutamine synthetase; Gase, glutaminase; GDH1, glutamate dehydrogenase (NADPH); AMPD, AMP deaminase; AlaT1, alanine transaminase 1; AlaT2, alanine transaminase 2; BCT1, branched-chain amino acid transaminase 1; BCT2, branched-chain amino acid transaminase 2. Statistical analysis (2-way anova) of the differences between groups. The variable “sex” showed global differences for Gase (P < 0.0001), eNOS (P = 0.0014) and AlaT2 (P = 0.0018). The variable “site” showed significant differences for all genes (P < 0.0001 for eNOS, Gase, GDH1, AMPD, BCT1 and AlaT2; P = 0.0005 for Ase1; P = 0.0014 for GS; P = 0.0023 for AlaT1, P = 0.024 for BCT2, and P = 0.039 for GCS) except N-acetyl-glutamate synthase. The application of post-hoc Tuckey test between male/female pairs are shown in the Figure as red stars between the corresponding columns.

In all sites, the expression of e-NOS was, at least one order of magnitude higher than arginase; subcutaneous WAT being an exception: despite showing a similar pattern of expressions, the levels of mRNA per g of tissue protein were higher for most genes, in subcutaneous WAT, than in the other three sites. There was a generalized predominance of glutamine synthetase expression over that of glutaminase. The glycine cleavage system (specifically the H protein of the complex) and AMP deaminase showed also a robust expression, at levels comparable to those of alanine transaminases. The two branched-chain amino acid transaminases were also within this range, but the expression of the form 2 was much higher.

The statistical comparisons of the data in Fig. 3 present limited effects for sex; overall only nitric oxide synthase, alanine transaminase 2 and glutaminase showed significant overall differences between female and male rats. Paired sex-related differences were concentrated in subcutaneous WAT, with higher male values in the expression of nitric oxide synthase, glutamine synthase (but female-predominant glutaminase), AMP deaminase and alanine transaminase 1. No sex-related differences were found in the other sites, except higher male values in alanine transaminase 2 of perigonadal WAT. The differences between sites, however, were more marked, affecting all genes studied except N-acetyl-glutamate synthase (low expression) and arginase 2, which was expressed only in subcutaneous WAT.

Gene expressions of proteins involved in WAT acyl-glycerol metabolism

Figure 4 presents the gene expressions of the key transporter and enzymes that regulate the lipogenic process from glucose to acetyl-CoA and from that metabolite to acyl-CoA, including the three most important WAT lipases. The data are presented in a log scale and the numerical data are shown in Table S4. In spite of a considerable uniformity in the patterns for all four sites, there were marked differences in the extent of gene expression. In general, subcutaneous WAT values were higher, than those of the other sites. Again, overall (for mesenteric, retroperitoneal and perigonadal WAT), female expression values tended to be higher than those of the males for genes coding proteins favoring glucose incorporation (GLUT4, hexokinase), generation of NADPH (glucose-6P dehydrogenase, malic enzyme), and lipogenesis (citrate: ATP lyase, and acetyl-CoA carboxylase). Regulation of pyruvate dehydrogenase by its inhibiting kinases was higher in males, suggesting a lower mitochondrial availability of acetyl-CoA. In males, mitochondrial handling of fatty acids (carnitine palmitoleoyl-transferases, acyl-CoA dehydrogenases) and lipolysis (except adipose triacylglycerol lipase) showed higher relative expressions than in females.

Figure 4 Expression of genes coding for transporter and enzymes related to lipogenesis from glucose and catabolism of lipid stores in WAT sites of male and female rats.

All data are the mean ± sem of 6 animals, and are presented in a log scale. The numerical data are shown in Table S4. Blue (dark colour) columns represent the males, and green (light colour) represents the females. GLUT4, glucose transpoirter 4; HK, hexokinase; G6PDH, glucose-6P dehydrogenase; ME, malic enzyme; PDHK2, pyruvate dehydrohenase kinase 2; PDHK4, pyruvate dehydrohenase kinase 4; CATPL, citrate: ATP lyase; AcCoAC, acetyl-CoA carboxylase; FAS, fatty acid synthase; PCATl, palmitoleoyl-carnitine acyl-transferase (liver); PCATm, palmitoleoyl-carnitine acyl-transferase (muscle); AcADH, acyl-CoA dehydrogenase; ATL, adipose triacylglycerol lipase; HSL, hormone-sensitive lipase; LPL, lipoprotein lipase. Statistical analysis (2-way anova) of the differences between groups. The variable “sex” showed global differences for HK (P = 0.0009), AcCoAC (P = 0.0035), GLUT4 (P = 0.0040), CATPL (P = 0.010), G6PDH (P = 0.020), PHDK4 (P = 0.020), ME (P = 0.024) and ATL (P = 0.026). The variable “site” showed significant differences for HSL, ATL and LPL (P < 0.0001), PCATl (P = 0.0004), G6PDH (P = 0.0005), PDHK2 (P = 0.0013). PDHK4 (P = 0.0016), ACADH (0.0022), HK (P = 0.0057), PCATm (P = 0.015) and AcCoAC (P = 0.0384). The application of post-hoc Tuckey test between male/female pairs are shown in the Figure as red stars (P < 0.05).

Subcutaneous WAT showed higher expression values for males in pyruvate dehydrogenase kinase 4, palmitoleoyl.carnitine acyl-transferase (liver), acyl-CoA dehydrogenase and both lipoprotein and adipose triacylglycerol lipases. In mesenteric WAT, the only significant difference was for higher malic enzyme expression in females. In retroperitoneal WAT, female expression values were higher for malic enzyme, citrate: ATP lyase and acetyl-CoA carboxylase. Again, in perigonadal WAT, female expression values were higher for glucose-6P dehydrogenase. The overall differences for “site” were significant for all genes investigated except for GLUT4, malic enzyme, citrate: ATP lyase and fatty acid synthase.

Comparison of female and male gene expression

Table 4 summarizes the sex-related differences in gene expression. The data are presented in two cooperative forms: expression per unit of protein and per unit of RNA weight in the tissue. The corresponding numerical data for RNA data and complete statistical analysis are shown on Table S5. Table 4 shows only the cases where female–male differences were significant, and the genes are divided in four sections. In the first, corresponding to urea-cycle enzymes, only carbamoyl-P synthase 2 of subcutaneous WAT showed higher female than male values. No other differences were seen. There was a high degree of superimposition between the data obtained from protein and RNA; but the number of enzymes with statistically significant between sexes was higher in most WAT sites when the data of reference was RNA than when related to tissue protein.

Table 4 Comparison of male-female specific expression of genes in different WAT sites with respect to tissue total protein or RNA.

The data are the mean ± sem of 6 animals per group, and are expressed as fmol of the corresponding gene mRNA per g of protein or mg of total RNA. The complete numerical data Table for RNA is presented in Table S5. Only significant differences are shown. M > F represents that male data were significantly higher than those of females; F > M represented that female data were significantly higher than those of males. Analysis of significance was done using 1- and 2-way anovas (the latter for combined sites). The data in regular font correspond to significant values in the expression of fmol/mg RNA, those in italics correspond to the data which were significant only when expressed as fmol/g protein. The data in bold correspond to differences statistically significant both when referred to tissue protein and RNA.

Parameter	WAT site	All sites P	
	SC	ME	PG	RP		
urea cycle enzymes	
Carbamoyl-P synthase 2	F > M	–	–	–	F>M	
Ornithine carbamoyl-transferase	–	–	–	–	–	
Argininosuccinate synthase	–	–	–	–	–	
Argininosuccinate lyase	–	–	–	–	–	
Arginase 1	–	–	–	–	–	
other enzymes of amino acid metabolism	
N-acetyl-glutamate synthase	M>F	–	–	–	–	
Glutamate dehydrogenase 1	–	–	–	–	–	
Glutamine synthetase	M>F	–	–	–	M>F	
Glutaminase	F>M	F>M	–	–	F>M	
AMP deaminase	M > F	–	–	–	M>F	
Glycine cleavage system	–	–	F>M	–	–	
Alanine transaminase 1	M > F	–	–	M>F	M>F	
Alanine transaminase 2	M>F	–	M > F	M>F	M > F	
Branched-chain amino acid transaminase 1	–	–	M>F	M>F	M>F	
Branched-chain amino acid transaminase 2	–	–	–	–	–	
Endothelial nitric oxide synthase	M > F	–	–	M>F	M > F	
enzymes (and transporters) related with lipogenesis from glucose	
Glucose transporter 4	–	–	–	F>M	F > M	
Hexokinase 2	–	–	F>M	F>M	F > M	
Glucose-6P dehydrogenase	–	F>M	F>M	F>M	F > M	
Malic enzyme	–	F > M	F>M	F > M	F > M	
Pyruvate dehydrogenase kinase 2	M>F	–	–	M>F	M>F	
Pyruvate dehydrogenase kinase 4	M > F	–	–	M>F	M > F	
Citrate: ATP lyase	–	F>M	–	F > M	F > M	
Acetyl-CoA carboxylase	F>M	F>M	–	F > M	F > M	
Fatty acid synthase	–	F>M	F>M	F>M	F>M	
enzymes (and transporters) related with lipolysis and fatty acid oxidation	
Carnitine palmitoleoyl transferase (liver)	M > F	–	M>F	M>F	M>F	
Carnitine palmitoleoyl transferase (muscle)	–	–	–	–	M>F	
Long-chain acyl-CoA dehydrogenase	M > F	F>M	–	–	M>F	
Adipose tissue triacylglycerol lipase	F > F	–	–	M>F	M > F	
Hormone-sensitive lipase	–	–	–	–	M>F	
Lipoprotein lipase	M > F	–	–	M>F	–	

Other amino acid metabolism data showed a relative predominance of higher relative expressions in males, especially affecting the transaminases. The sites with more differences were subcutaneous and retroperitoneal WAT. In contrast, lipogenesis was more highly expressed in females, with minimal effects on subcutaneous and highest in retroperitoneal WAT. Male predominance was observed again on pyruvate dehydrogenase kinases’ expressions, which increase marks a lower rate of production of acetyl-CoA as substrate for acyl-CoA synthesis. Finally, in the analysis of lipolytic and lipid oxidation-related genes, the male higher values were the norm, especially in subcutaneous and retroperitoneal WAT.

DISCUSSION

The results presented support a wide extension of amino acid metabolism in different sites of WAT, with enzyme activities and expressions following similar patterns in all four sites studied. In addition to urea cycle, AMP-deaminase (Arola et al., 1981a), glutamine synthetase (Arola et al., 1981b), glutamate dehydrogenase (Arola et al., 1979) and nitric oxide synthase (Pilon, Penfornis & Marette, 2000), we found that WAT expresses the glycine cleavage system (at least the H protein), so far not described.

The metabolic capabilities of WAT with respect to amino acid metabolism are probably more extensive than usually assumed (Alemany, 2012b), largely because it is unknown, with scant literature references to WAT amino acid metabolism (López-Soriano & Alemany, 1986; Kowalski, Wu & Watford, 1997; Herman et al., 2010; Lackey et al., 2013). The range of expressions observed for amino acid metabolism-related enzymes in the four WAT sites studied (Figs. 2 and 3) was mostly in the 5–500 fmol/g protein. In comparison, the expressions (Fig. 4) for lipogenesis, the (assumed) main metabolic function of WAT, and other lipid metabolism-related expressions were in the range of 10-1,000 fmol/g protein. Thus, the differences between lipogenesis and amino acid metabolism-related gene expressions were not as extensive as expected from the known massive mobilization/ deposition of triacylglycerols in adipocytes, compared with the relatively low level of cell proteins (Salans & Dougherty, 1971) (and cytoplasm) of WAT. This is compounded by the lack of sufficient data on WAT amino acid metabolism above indicated. The relatively elevated amino acid metabolism enzyme levels and gene expressions found hint at a potential relative importance of WAT on body amino acid metabolism.

The considerable uniformity of WAT urea cycle-enzyme activities and expressions, and their marked independence of sex can be interpreted essentially in two ways: (a) as playing a minimal metabolic role: i.e., a residual, secondary or specialized pathway. Or, alternatively, (b), it can be assumed to be a consequence of a well-established and robust homeostatic maintenance of its function. That is, a role critical enough not to be sensibly influenced by external regulatory factors such as sex hormones. The first possibility may seem the more obvious, but it is insufficient to counter a number of critical arguments: first of all, the unexpectedly high level of enzyme expressions and activities. Individual urea cycle enzymes are present in many tissues, Emmanuel (1980), Nishibe (1974), and Rath et al. (2014). However, the mere existence of a complete urea cycle in a peripheral organ outside the splanchnic bed has not been previously described, as far as we know. A full operative urea cycle has been described only for liver (Emmanuel, 1980). The key question is whether this cycle is functional or not. Due to its methodological difficulties, it should be studied using other (i.e., tracer) techniques. However, the relatively high enzyme activities and gene expressions observed, and the fact that all urea cycle and related ammonium metabolism enzymes are present suggest that this distribution has a clear functional purpose. This is, consequently, a situation different from that of tissues, which contain only part of the cycle to serve other metabolic purposes. The lack of sex-related differences in WAT sites of urea cycle compared with lipogenic processes, as shown in Figs. 2–4 and the remarkable uniformity in pattern distribution between WAT sites also support the functionality of WAT urea cycle.

The varying ratios of activity/expression suggest a main post-translational control, extended to all sites. The high ornithine carbamoyl-transferase vs. arginino-succinate synthase activities suggest a probable implication in the peripheral (and critical) synthesis of citrulline (Yu et al., 1996), which may complement its conversion by the kidney (Borsook & Dubnoff, 1941). WAT participates in substrate cycles, including alanine synthesis (Snell & Duff, 1977) and glutamine release (Kowalski & Watford, 1994) in which amino acids are implicated. This analysis is further complicated by the quantitative importance of both activity and gene expressions compared with those of lipogenesis, the mainstay of WAT metabolism. Our understanding of these differences is complicated by the overall large size of the adipose organ (Romero et al., 2014), even taking into account the metabolically inert mass of fat. Taken together, these arguments support a significant role of WAT in amino acid metabolism.

Our data suggest, in any case, a clear site-sex interaction (Lemonnier, 1972; Jaubert et al., 1995) that brings up differences in the expression of several amino acid metabolism-related genes other than urea cycle, which remains uncannily undisturbed and globally uniform. In males, subcutaneous WAT shows higher expressions for genes related to transfer of acyl-CoA to the mitochondria and its oxidation than females; this is consistent with the possibility of using fatty acids as energy substrate. The higher male inhibition of pyruvate dehydrogenase, by kinases, generalized to most sites, reinforces this trend. On the other side, the expression of WAT lipogenic enzyme genes (especially when expressed with respect to total RNA) was higher in females than in males. In contrast, lipases practically did not show differences. The different female-male specific metabolic predominance in WAT sites prove that the limited changes in urea cycle, compared with lipogenesis, could not be solely a consequence of overall lack of effects of sex on WAT; but it is, instead, a specific characteristic of WAT urea cycle as compared to other metabolic pathways.

The higher lipogenic (and lower lipolytic) gene expressions of female WAT is counterintuitive when we think of the higher WAT mass of adult male rats (Romero et al., 2014). Probably, the lower male gene expressions found here mirror a less active metabolism, in conjunction with the intestine and liver. The visceral fat accumulation in adult men (Bosch et al., 2015) as compared with women is correlated with insulin resistance (Pascot et al., 2000) and other metabolic syndrome-related pathologies (Watanabe & Tochikubo, 2003). The discordances in sex-related control of metabolism and fat deposition between humans and rats are a critical caveat against generalization to humans of what is found using animal models, in spite of shared mechanisms and trends.

In addition to the human-rodent question, the main limitation of this study is the lack of previous data with which establish comparisons, made even more difficult by our scarce knowledge of amino acid metabolism. The extensive and interconnected net of pathways needs to be investigated. The most critical handicap, however, is the lack of a critical mass of scientists and of actualized methodology: specific protein measurement reagents (antibodies), and/or methods (and products) for the estimation of enzyme activities and metabolites of amino acid metabolism. Consequently, the data we present here should be taken as just an initial foray into a highly promising field of study.

An additional question may help explain the differences between sites (Lemonnier, 1972; Rydén et al., 2014), and the influence of sex (and sex hormones), the differences in cell populations of different WAT sites. A factor that affects the mean adipocyte size (also influenced by obesity (Garaulet et al., 2006)) and the presence of other types of cells, in different proportions, such as macrophages (Králová Lesná et al., 2015), stem cells (Ogura et al., 2014) and other stromal components (Maumus et al., 2011).

Notwithstanding these caveats, the data gathered all point to a few preliminary conclusions, which could not be yet fully proven with the data we presented, largely because no other results are available for comparison or independent confirmation. The potential for lipid handling of WAT sites was strongly modulated by sex, being considerably dependent on the site studied. This part of the study, devised to provide a background comparison for amino acid metabolism showed more extensive differences than expected, and needs to be studied more specifically and deeply before sufficiently based conclusions could be extracted.

There was a considerable stability of the urea cycle activities and expressions, irrespective of sex, and with only limited influence of site. Which we interpret as this cycle operation being more general than the specialized site metabolic peculiarities, with robust control of WAT urea cycle, probably related to a possible role as provider of arginine/ citrulline (Beliveau Carey et al., 1993). The resilience to change of urea cycle, in the context of a tissue characterized by its plastic adaptability, supports a generalized, probably essential, role in overall amino N handling.

In contrast, sex affected deeply WAT ammonium-centered amino N metabolism in a site-related fashion, with relatively higher levels of activity in males and in female subcutaneous WAT. The data on amino acid catabolism fit also with a role of mesenteric WAT as gatekeeper of the portal system, the hypothesis advanced for WAT glucose disposal (Arriarán et al., 2015) can be easily translated to the management of possible transient excesses of dietary amino acids.

In sum, WAT seems to play significant role in overall amino acid metabolism, including a functional urea cycle, which is not affected by sex. Contrary to lipid/glucose-related pathways, the data presented point to a centralized control of urea cycle operation affecting the adipose organ as a whole.

Supplemental Information

Supplemental Information 1 Supplemental material

Click here for additional data file.

Additional Information and Declarations

Competing Interests

Author Contributions

Animal Ethics

Data Availability

The authors declare there are no competing interests.

Sofía Arriarán performed the experiments, analyzed the data, prepared figures and/or tables, reviewed drafts of the paper.

Silvia Agnelli performed the experiments.

Xavier Remesar analyzed the data, contributed reagents/materials/analysis tools, reviewed drafts of the paper.

José Antonio Fernández-López analyzed the data, prepared figures and/or tables, reviewed drafts of the paper.

Marià Alemany conceived and designed the experiments, wrote the paper, prepared figures and/or tables, reviewed drafts of the paper.

The following information was supplied relating to ethical approvals (i.e., approving body and any reference numbers):

Committee on Animal Experimentation of the University of Barcelona, Authorization: DMAH-5484

The following information was supplied regarding data availability:

University of Barcelona, CRAI Repository;

http://hdl.handle.net/2445/66872.

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
