# Peer review of "Effects of sex and site on amino acid metabolism enzyme gene expression and activity in rat white adipose tissue"

_PeerJ, doi:10.7717/peerj.1399_

## Round 0.1 · original submission · Major Revisions

· Academic Editor

Major Revisions

Although both reviewers note that the data presented are potentially novel and interesting, and the experiments seem appropriate, they also note that the manuscript requires extensive revision before it could be considered acceptable for publication.

As also noted by both reviewers, the manuscript requires extensive English Language editing, and the use of a professional English Language editing service, or significant input from a native English speaker, is recommended.

In addition to the reviewers' comments:

Abstract:
* Line 38: In addition to reviewer 1's comment, I think that assumed is not an appropriate term - perhaps consider "reported" instead.

Introduction:
* Line 59: "These differences are often overlooked when analyzing the regulation and metabolic responses of adipose tissue, especially under conditions of inflammation." Appropriate references should be included for this statement.
* Line 63: "metabolic peculiarities of adipose tissue" is an unusual statement. Why are these peculiar? Do you mean metabolic processed that are specific to adipose tissue?
* Line 65: "a deep reinterpretation" - perhaps a term such as comprehensive would be more appropriate than deep.
* Line 72: "has been largely neglected" - this statement should have supporting references cited. There appear to be several studies which specifically investigate amino acid metabolism in adipose tissue. Further description of this point seems warranted.
* Line 87: "Women and female rats alike, show a powerful resistance to fattening" - this statement seems at odds with the increasing rates of obesity observed in both men and women - I think some further explanation is required.

Materials and methods:
* If available, an ethics approval number should be included.
* What evidence prior to this study did the authors have that n=6 per group would be an appropriate number of animals? Was a power calculation performed? If so, this should be included.
* Line 121: "Lactate was measured with" - there seems to be some information missing here.

Discussion:
* Line 281: "The differences between lipogenesis and amino acid metabolism-related gene expressions were not as extensive as expected," What was this expectation based on? If on previously published data, this should be cited. If not, why was there an expectation of smaller differences?

Reviewer 1 ·

Basic reporting

It the present manuscript Arriarán et al study amino acid metabolism and processes related to glucose and lipid metabolism in adipose tissue depots in male and female rats fed a standard chow diet. The main readout is gene expression levels. In addition to qPCR data adipose tissue sizes, concentration of some blood metabolites and activity of urea cycle enzymes are presented.
The manuscript presents some novel data on gene expression and identifies depot differences and sex dimorphism of potential interest. The experiments also appear to have been performed in a correct way. However, the manuscript is poorly written, the aims are quite unclear, important controls are lacking and the interpretation of data is questionable. Therefore I think that the manuscript cannot be published in its present form. Below I will list a number of comments on specific issues in the manuscript that needs to be addressed. The list of text segments that needs to be improved is extensive and my comments only covers some of the problems. The manuscript needs to be shortened and cleaned up, both when it comes to content and language. I think that the authors should consider consulting an (English speaking) editor.

Comments:
22 Standard chow diet is not obesogenic.
23 misspelling of urea
21-23 objective not clear and specific
31 Speculative sentence
32 probably essential is speculative
33 I wouldn’t say deeply
37 unclear
39 usually assumed: where, by who
47-72 Most of this section is not relevant for the present study
64 what do you mean?
68-69 strange sentences
74 Please show unpublished results
75 inter-organ N handling?
83 What interest? Your interest? If there are few published studies I would say that there is limited interest in the field. Sounds like a paradox to me.
85 explain the relevance of difference in consumption for the present study
87-88 a weird sentence
109 were the mice fasted. When do you consider the light cycle to start. This is important for the interpretation of the data.
172 Describe the experiment it the result section, it is difficult to read this way
177 The authors should perform control experiments to check for tissue composition. Differences in protein and RNA concentration imply that there may be other cell types/organs present in the samples. For example, subcutaneous fat often contains beige fat and therefore brown adipose tissue markers should be used. Mesenteric fat is full of lymph nodes and blood vessels. Markers for these organs should be used. The presence of these tissue types and their distribution between the adipose tissue depots should be taken into consideration in the analysis of the data. Especially the comparison of the tissue depots may be affected by this.
180 Why show glucose data if it is not correct
206 What is the purpose of comparing expression of different genes when nothing is known about the kinetics of the enzymes, post transcriptional regulation and so on. (this comment also applies for several segments in the manuscript, it is for example important for the discussion regarding quantitative differences between amino acid metabolism and metabolism of glucose and lipids)
226 Check for beige fat, see above
231 if it is not significant you should not discuss difference
230-234 Difficult to understand
244-250 Table 4 does not fit into the paper and should be removed. I do not understand why the authors change from protein to RNA for normalization. If there is no particular reason to compare RNA and protein normalization the authors need to be consistent throughout the paper. It is also unscientific to use randomly chosen cut off values instead of proper statistics. Changes in expression should only be discussed for genes where there is a true statistical difference. The discussion based on the results in Table 4 needs to be adjusted accordingly.
277-278 Motivate and add references
282-283 Motivate and add references
288-302 Some of these arguments are merely speculations
301-302 Reference
315-316 Compare to the present data and to data from rodents
323-326 There are several open access microarray dataset available at GEO and Arrayexpress comparing gene expression in adipose tissue depots. The authors should use these to put the data into a context. Also some papers specifically studying adipose tissue depot dependent differences in metabolism in rodents (e.g. Caesar et al 2010) and humans (e.g. Tchernof et al 2006) have been performed and should be referred to.
Figure legend 2-4 Spell out the full names of the genes
Table 2 cannot find superscript letters
Table 2 Text in two last lines makes no sense
Table 3 What does “main energy plasma parameters” actually means
Table 3 * at the end of the text
Table 3 Wrong statistical method, use t-test and FDR
Table 4 Should be removed (see above)

Experimental design

see "Basic reporting" section

Validity of the findings

see "Basic reporting" section

Additional comments

see "Basic reporting" section

Reviewer 2 ·

Basic reporting

Arriaran et al present a manuscript detailing the site and sex specific changes in markers of amino acid metabolism in adipose tissue. While the scope of this manuscript is interesting, there are numerous typographical and grammatical errors, the manuscript may benefit from review by a native english speaker. However the content itself is appropriate. The authors state in the abstract that metabolic studies on the WAT, particularly amino acid metabolism is considerably limited. This should be rephrased as there is a range of papers on amino acid metabolism in adipose tissue in the literature.

Experimental design

The experimental design is appropriate, however I have several questions?
1. Were the female rats staged, if so what stage of the estrus cycle were they culled at?
2. The statistics section is incomplete, mention post-hocs, if SEM or SE was used etc.
3. More that one reference gene should be included as part of the RT-PCR analysis
4. In table 2 each depot should be presented as both g (as in the paper) and also %BW (it seems that the %BW for each depot is similar between both males and female). If the females are 150ish grams lighter than the males of course the tissue weight will be different, the proportion of fat in these depots is a much more important measure.
5. I'm not sure how relevant the differences in RNA and protein proportions are? It seems like a crude measure.
6. The fonts in table 1 should be consistent throughout.

Validity of the findings

How do the authors reconcile the differences between gene expression and enzyme activity in Figure 2?
Figure legends should not be used to summarize the results.
In Figure 4 there are pretty big differences between males and females for a range of genes (GLUT4, CATPL, ACoAC, FAS etc), and the error bars appear to be quite neat, are you sure they are not significant?

---

## Round 0.2 · Minor Revisions

· Academic Editor

Minor Revisions

As both reviewers have noted, you have constructively incorporated the review comments into the revised version, resulting in a significantly improved manuscript. You have also adequately addressed all of my comments regarding the original manuscript.

The main point still to address is that of normalization raised by Reviewer 1. In addition, you should consider including a comment regarding WAT cell composition in the discussion (as raised by Reviewer 1), and undertake a final careful check of the manuscript for typographical errors as noted by Reviewer 2.

Reviewer 1 ·

Basic reporting

In the new version of the manuscript by Arriarán et al the authors have greatly improved the readability and the focus of the text. Many of the problems that I commented on in my reveiew of the manuscript have been dealt with.
My main remaining criticism of the manuscript is still the change of normalization between the figures and Table 4. As the authors point out tissue weight and cell number are not suitable factors for normalization here, but why shift from proteins to RNA? In my opinion the motivation given in the newly introduced paragraph in the method section is not strong enough and the dual way of presenting data confuses more than helps in the interpretation of the results.
Other remarks:
I realize that an analysis of cell types in WAT is out of the scope of this manuscript but I think that differences in cell composition should be mentioned in the discussion.

Experimental design

no comments

Validity of the findings

no comments

Additional comments

no comments

Reviewer 2 ·

Basic reporting

The authors have taken on board comments and made significant attempts to amend the manuscript

Experimental design

The design of the experiments is appropriate

Validity of the findings

The findings are valid and add the literature on adipose tissue biology

Additional comments

Overall the manuscript is much improved. There are still some minor typos throughout so authors should go through thoroughly.

---

## Round 0.3 · accepted · Accept

· Academic Editor

Accept

As you will see from Reviewer 1's comments below, there are still some concerns regarding the normalization procedure used. However, as also pointed out by this reviewer, the inclusion of both normalization methods in this version of the paper means that the reader will be able to assess the approach and evaluate the data on this basis.

I therefore feel that the manuscript is now suitable for publication.

Reviewer 1 ·

Basic reporting

Even though I still think that it is questionable to change normalization method to obtain significance the introduction of protein normalized statistics in italics in table 4 at least makes in more easy for the reader to evaluate the data.
I think that the manuscript now is suitable for publication.

Experimental design

see above

Validity of the findings

see above

Additional comments

see above